# Assessing medication use patterns in patients hospitalised with COVID-19: a retrospective study

Tanja Mueller ,[1,2] Amanj Kurdi ,[1,2,3] Elliott Hall,[2] Ian Bullard,[4] Jo Wapshott,[4] Anna Goodfellow,[4] Niketa Platt,[2,5] Euan Proud,[2] Stuart McTaggart ,[6] Marion Bennie ,[1,6] Aziz Sheikh,[7,8] on behalf of the EAVE II Collaboration

For numbered affiliations see end of article.

**Correspondence to**
Dr Tanja Mueller;
tanja.muller@strath.ac.uk

## ABSTRACT

**Objective** To describe patterns of medication use—that is, dexamethasone; remdesivir; and tocilizumab—in the management of patients hospitalised with COVID-19.

**Design and setting** Retrospective observational study, using routinely collected, linked electronic data from clinical practice in Scotland. Data on drug exposure in secondary care has been obtained from the Hospital Electronic Prescribing and Medicines Administration System.

**Participants** Patients being treated with the drugs of interest and hospitalised for COVID-19 between 1 March 2020 and 10 November 2021.

**Outcomes** Identification of patients subject to the treatments of interest; summary of patients' baseline characteristics; description of medication use patterns and treatment episodes. Analyses were descriptive in nature.

**Results** Overall, 4063 patients matching the inclusion criteria were identified in Scotland, with a median (IQR) age of 64 years (52–76). Among all patients, 81.4% (n=3307) and 17.8% (n=725) were treated with one or two medicines, respectively; dexamethasone monotherapy accounted for the majority (n=3094, 76.2%) followed by dexamethasone in combination with tocilizumab (n=530, 13.0%). Treatment patterns were variable over time but roughly followed the waves of COVID-19 infections; however, the different drugs were used to varying degrees during the study period.

The median (IQR) treatment duration differed by medicine: dexamethasone 5 days (2–9); remdesivir 5 days (2–5); and tocilizumab 1 day (1–1). The overall median (IQR) length of hospital stay among all patients included in the study cohort was 9 days (5–17); 24.7% of patients died in hospital.

**Conclusion** The use of adjuvant medicines in patients hospitalised with COVID-19 appears in line with evolving evidence and changing treatment guidelines. In-hospital electronic prescribing systems are a valuable source of information, providing detailed patient-level data on in-hospital drug use.

## INTRODUCTION

Similar to other countries globally, COVID-19 had a considerable impact on population health in the UK; in Scotland alone, approximately 44 300 hospital admissions and 12

## STRENGTHS AND LIMITATIONS OF THIS STUDY

⇒ This study used patient-level data collected as part of routine care.
⇒ Available data enabled an in-depth description of the use of repurposed drugs in patients hospitalised with COVID-19 in Scotland.
⇒ Due to the observational nature of this study and limitations on the data, analyses of clinical safety or effectiveness of the drugs of interest were not feasible.

500 deaths have been attributed to COVID-19 between March 2020 and December 2021.[1] Even though vaccines were rolled-out at speed, the search for effective treatments is ongoing due to limits on vaccine effectiveness,[2] immune escape particularly among variants of concern[3] and waning immunity following both infection and vaccination.[4] While a range of diverse options were discussed early in the pandemic—spanning from antibiotics to convalescent plasma— only a small number of medicines were found to be effective in clinical trials,[5] and subsequently approved for use by regulators. Medicines recommended for use in patients hospitalised with COVID-19 in the UK at the time of study conduct comprised remdesivir, an antiviral drug (introduced in guidelines on 29 May 2020); steroids, most prominently dexamethasone (16 June 2020); and monoclonal antibodies including tocilizumab (8 January 2021).[6–8] Guidelines have however undergone frequent changes over the last 2 years.[9] Dexamethasone is now routinely given to patients with low oxygen saturation, primarily because it is an effective and generally well tolerated anti-inflammatory drug, with extensive experience in usage. Remdesivir is recommended in patients receiving low-flow additional oxygen, depending on renal and liver function, to be given within

the first few days of infection as it is deemed ineffective at later stages; tocilizumab is recommended in patients receiving respiratory support.[10] While usually given separately, consecutive (eg, dexamethasone together with tocilizumab) or subsequent treatment with multiple drugs can be considered based on clinical condition.[10 11] Most recently, a number of antivirals and monoclonal antibodies have also been approved for use in the community, including, for example, molnupiravir and sotrovimab.[12]

There is a need for real-world studies assessing the uptake, effectiveness, and safety of these drugs to complement the findings from rapidly undertaken efficacy trials. Nevertheless, thus far, evidence from clinical practice is limited; in particular, there are very few publications offering detailed descriptions of in-hospital drug use in Scotland. While a range of studies have investigated the use of repurposed and adjuvant drugs in patients with COVID-19, the majority of the publications have focused on the effect of specific drugs on disease outcomes rather than the drug use itself.[13–15] Although there are exceptions,[16] studies rarely provide in-depth information on treatment patterns (including temporal and geographical variation); details on duration of treatment are even less common, possibly due to the unavailability of in-depth, patient-level information.[17–19]

In the UK, the drive towards digitisation in healthcare has resulted in the implementation of electronic prescribing; initially in primary care, but subsequently also in specialist settings (eg, oncology) and secondary care. In Scotland, the roll-out of the Hospital Electronic Prescribing and Medicines Administration (HEPMA) system was initiated in 2014; HEPMA is now available in hospitals across 6 of the 14 regional organisations (Health Boards) within NHS Scotland tasked with planning and delivering services to their local populations.

The aim of this study was to investigate the use of in-hospital medication for the treatment of COVID-19, namely dexamethasone, remdesivir, and tocilizumab. Our specific objectives were: to describe medicines use over time and across geographical areas; to summarise patients' baseline characteristics; and to describe treatment patterns as well as hospital admission episodes, including their outcomes.

## METHODS
### Study design and setting
The study was designed as a retrospective cohort study using routinely collected linked healthcare data, covering the time period from 1 March 2020 to 10 November 2021 based on the timeline of events (WHO interim guidance/declaration of the pandemic) and data availability. Healthcare throughout the UK is provided via the tax-funded National Health Services (NHS), with services being offered primarily without payment at the point of care.[20]

The study setting was Scotland, where every resident is allocated a unique patient identifier (the Community

Health Index (CHI) number) the use of which is mandatory throughout the health and care system,[21] thereby enabling deterministic linkage of electronic patient records. Since implementation of technical solutions takes place at Health Board level and regional time frames depend on local requirements and circumstances, at the time of study conduct, HEPMA had been implemented fully or partially in six Health Boards: NHS Ayrshire and Arran; NHS Dumfries and Galloway; NHS Forth Valley; NHS Greater Glasgow and Clyde (GGC); NHS Lanarkshire; and NHS Lothian, covering an estimated 3.6 million people or approximately 65% of the Scottish population (5.4 million).[22] However, as the HEPMA implementation started at different points in time in the various Health Boards and roll-out across hospital sites was incremental, data coverage was incomplete for two Health Boards: NHS GGC, where implementation started in late 2020, that is, after the start of the study period; and NHS Lothian, where very few hospitals were initially included. In addition, coverage within individual hospitals differed due to the use of existing systems instead of (or in addition to) HEPMA in certain areas, for example, in intensive care units (ICUs).

### Data sources
Briefly, HEPMA was used to extract any prescriptions of the medications of interest. Using CHI numbers, HEPMA data was subsequently linked to the Scottish Morbidity Records inpatient data set and the Electronic Communication of Surveillance Scotland data to identify the study cohort and ensure that the medications of interest were administered during a COVID-19-related hospital admission episode. A number of other data sets (including, but not limited to, primary care prescriptions and death records) were also available through the Early Pandemic Evaluation and Enhanced Surveillance of COVID-19 (EAVE II) surveillance platform.[23] Further details can be found in the protocol.[24]

### Study population
With drug use being the specific focus of the study, the main study cohort comprised patients who had received at least one of the medications of interest (ie, dexamethasone, remdesivir, tocilizumab) while being admitted to hospital in the aforementioned Scottish Health Boards between 1 March 2020 and 10 November 2021 for suspected or confirmed COVID-19. Hospitalisations for COVID-19 were defined as admissions within 28 days of a positive reverse transcriptase PCR (RT-PCR) test and/or with an International Classification of Diseases, 10th edition code for COVID-19 (U07.1, U07.2) recorded on a patient's hospital admission file as the primary or secondary diagnosis.

### Drug use, covariates and statistical analyses
The primary outcome was observed treatment patterns among patients with any of the medications of interest over the duration of the study period, including both

monotherapy as well as treatment with any combinations thereof. In addition, secondary outcomes included summaries of exposure in terms of administered dose and duration of treatment; as well as length of hospital stay and in-hospital mortality.

Patient characteristics, including socio-demographic factors,[25] COVID-19 vaccination status, disease-related aspects, comorbidities,[26] and concomitant medication,[27] were summarised at baseline. Detailed definitions, descriptions and relevant data sources can be found in the protocol.[24]

All analyses were descriptive in nature; results were expressed as counts/frequencies for categorical variables, and median/IQR for continuous variables. Patients' index dates—that is, the date of the first recorded prescription for any of the drugs of interest for each patient— were used to classify treatment over time. Analyses were conducted using R/R Studio V.3.6.1, and Stata V.15.

## Patient and public involvement

The EAVE II Public Advisory Group is a diverse group of patient and public involvement (PPI) contributors who meet monthly to incorporate the views of patients and the public into research using the EAVE II data set. This includes shaping of research via the EAVE II Steering Group, which is attended by two lay leads. A lay summary, co-written by our PPI contributors, will be shared via the outputs section of the EAVE II website, hosted by the University of Edinburgh.[23]

Reporting of the study follows the STROBE (Strengthening the Reporting of Observational Studies in Epidemiology) guidelines[28]; a STROBE checklist can be found in the online supplemental material.

## RESULTS

At the study locations in Scotland, 4063 patients admitted to hospital with a suspected or confirmed diagnosis of COVID-19 received at least one of the drugs of interest during the study period. Of these, 3745 patients (92.2%) had COVID-19 recorded as their primary diagnosis for hospital admission; 109 patients (2.7%) did not have COVID-19 as a diagnosis on their admission records and were identified based on a positive RT-PCR test alone. The vast majority (n=3913, 96.8%) of admissions were emergency admissions, with the majority admitted via the acute assessment unit significant care facility (n=2157/4063, 53.1%).

The largest group of patients was treated in NHS Lanarkshire (n=1353, 33.3%), followed by NHS Ayrshire and Arran (n=868, 21.4%); NHS Forth Valley (n=758, 18.7%); and NHS GGC (n=614, 15.1%). The remaining two Health Boards—NHS Dumfries and Galloway and NHS Lothian—accounted for relatively small patient populations (n=248 (6.1%) and n=222 (5.5%), respectively).

## Patient baseline characteristics

The median age of all patients at the time of hospital admission was 64 years (IQR 52–76), with 37.6% of the patients over 70 years of age. Just over half of the patients (55.6%) were men. While approximately half of all patients (n=2159, 53.1%) had a Charlson score of 0 indicating no comorbidities requiring hospitalisation during the 5-year period prior to their COVID-19 admission, the majority (n=3616, 89.0%) had received at least one prescription drug in primary care in the preceding 6 months. The most commonly prescribed medication included antihypertensive drugs, statins and antithrombotic drugs; polypharmacy—that is, being prescribed five or more different medications simultaneously—was observed among 2752 patients (67.7%). See table 1 for details, and online supplemental tables S1 and S2 in the online supplemental material for a breakdown by individual drug.

## Medication use patterns

Treatment with dexamethasone, remdesivir and tocilizumab in patients with COVID-19 was variable over time. Generally speaking, prescribing patterns roughly followed the waves of COVID-19 infections; however, the different medicines were used to varying degrees and at diverging points in time during the study period. While dexamethasone use was observed throughout this period, remdesivir was primarily prescribed in 2020, whereas routine tocilizumab use started in early 2021; see also figure 1, and online supplemental figure S1 in the supplementary material for a representation of proportional changes in drug use over time. For further context, figure 2 depicts COVID-19 hospitalisations in Scotland during the study period.[29]

Furthermore, although the majority of patients were treated with one drug only (n=3307, 81.4%), some patients received two (n=725, 17.8%) or all three (n=31, 0.8%). Most of the patients treated with more than one drug received dexamethasone and tocilizumab in combination (n=530/756, 70.1%), followed by dexamethasone and remdesivir (n=185, 24.5%). While the vast majority of the patients with multiple drugs (n=748/756, 98.9%) received those during the same hospital admission episode, overall, the timing/sequencing of the drugs given was very diverse, with no clear patterns emerging among the patients who received two or more of them. For details, see table 2.

## Treatment episodes

Patients usually started treatment very soon after admission, with a median time between hospital admission and treatment initiation of 1 day (IQR 1–2) across the entire Scottish cohort; there was very little difference with regards to treatment initiation between the individual drugs or any of the combinations thereof. The median treatment duration overall was 4 days (IQR 1–8) but this differed by drug and whether patients were treated with one drug only, or a combination of drugs: individually,

**Table 1**  Patient baseline characteristics in Scotland

| n (%) | Scotland n=**4063** |
|---|---|
| Socio-demographics | |
| Median age (IQR) (years) | 64 (52–76) |
| Age 18–40 (years) | 504 (12.4) |
| Age 41–70 (years) | 2033 (50.0) |
| Age >70 (years) | 1526 (37.6) |
| Sex (male) | 2260 (55.6) |
| Most deprived (quintile)* | 1227 (30.2) |
| Least deprived (quintile)* | 383 (9.4) |
| Vaccination status | |
| Unvaccinated | 2964 (73.0) |
| One dose | 172 (4.2) |
| Two doses | 927 (22.8) |
| Charlson score† | |
| 0 | 2159 (53.1) |
| 1–2 | 1339 (33.0) |
| 3–4 | 383 (9.4) |
| >4 | 182 (4.5) |
| Medication at baseline: number of different items‡ | |
| 0 | 447 (11.0) |
| 1–4 | 864 (21.3) |
| 5–10 | 1339 (33.0) |
| >10 | 1413 (34.8) |
| Medication at baseline: specific drug classes (yes)‡ | |
| Antihypertensive drugs | 2141 (52.7) |
| Antithrombotic drugs | 1348 (33.2) |
| Statins | 1511 (37.2) |
| Bronchodilator | 984 (24.2) |
| Inhaled steroid | 814 (20.0) |
| Diabetes medication | 778 (19.2) |

*Deprivation based on the Scottish Index of Multiple Deprivation quintiles, where 1=most deprived and 5=least deprived.[25]
†Charlson score at baseline based on diagnoses from hospital discharge records during the 5-year period directly preceding the admission, identified using International Classification of Diseases,10th edition codes.[26]
‡Polypharmacy and medication at baseline based on prescriptions dispensed in community pharmacy during the 6 months period directly preceding the admission, identified using British National Formulary (BNF) codes[27]; includes both acute and chronic medication.

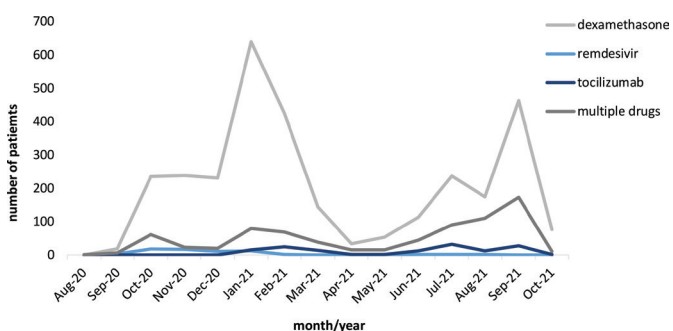

**Figure 1**  Prescribing of drugs of interest over time in Scotland, August 2020 to October 2021.

based on patient weight, which was reflected in the observed first/daily and overall doses given to patients. For dexamethasone, the median dose was 6 mg one time per day (IQR 6–6), with a median total dose of 30 mg (IQR 12–48); however, the median total dose was lower in patients treated exclusively with dexamethasone (24 mg, IQR 12–48) in line with the shorter duration of treatment in comparison to patients treated with a combination of drugs. For remdesivir, the starting dose was 200 mg at day one followed by 100 mg one time per day on subsequent days for the vast majority of patients, with an overall median total dose of 600 mg (IQR 300–600). The median dose of tocilizumab given to patients in Scotland was 800 mg (IQR 600–800).

The overall median length of hospital stay of all patients included in the study cohort, regardless of whether patients were discharged or died, was 9 days (IQR 5–17), with minor differences between patients being treated with dexamethasone (median 8 days, IQR 5–17); remdesivir (median 11 days, IQR 8–18); or tocilizumab (median 13 days, IQR 8–24) as a single agent. The majority of the patients (2807, 69.1%) were discharged to a private residence, whereas 1002 patients (24.7%) died in hospital.

## DISCUSSION

Using routinely collected data, our findings provide a picture of clinical practice during the study period in Scotland and demonstrate differences in the use of specific drugs for the treatment of COVID-19, as well as changes

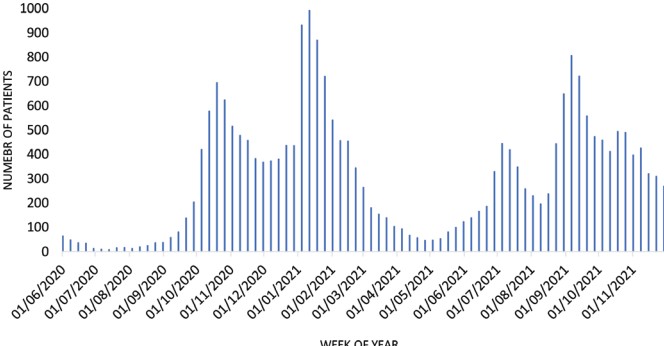

**Figure 2**  COVID-19 hospitalisations in Scotland, June 2020 to November 2021.

dexamethasone and remdesivir were given for a median of 5 days (IQR 2–8 and 3–5, respectively); in contrast, among patients treated with dexamethasone and one other drug (either remdesivir or tocilizumab), median duration of dexamethasone treatment was 7 days (IQR 4–9). Tocilizumab was given as a single dose only (median 1 day, IQR 1–1) regardless of treatment modality.

While standard dosing regimens were in place for both dexamethasone and remdesivir, tocilizumab is dosed

**Table 2** Patterns of drug use in Scotland between 1 March 2020 and 10 November 2021

| n (%) | Overall* | Dexamethasone† | Remdesivir† | Tocilizumab† | More than one drug‡ |
|---|---|---|---|---|---|
| Scotland | | | | | |
| Total | 4063 | 3094 (76.2) | 68 (1.7) | 145 (3.6) | 756 (18.5) |
| March–June 2020 | 6 | 6 (100) | 0 | 0 | 0 |
| July–September 2020 | 31 | 21 (67.7) | 3 (9.7) | 0 | 7 (22.6) |
| October–December 2020 | 860 | 711 (82.7) | 45 (5.2) | 0 | 104 (12.1) |
| January–March 2021 | 1461 | 1203 (82.3) | 15 (1.0) | 55 (3.8) | 188 (12.9) |
| April–June 2021 | 290 | 199 (68.8) | 1 (0.3) | 16 (5.5) | 74 (25.5) |
| July–September 2021 | 1324 | 874 (66.0) | 4 (0.3) | 73 (5.5) | 373 (28.2) |
| October–November 2021§ | 91 | 80 (87.9) | 0 | 1 (1.1) | 10 (11.0) |

*Includes all patients treated with at least one of the drugs of interest during the study period; dates of inclusion of each drug in National Health Services treatment guidelines: remdesivir 29 May 2020; dexamethasone 16 June 2020; tocilizumab 8 January 2021.[6–8]
†Includes only patients treated exclusively with one of the drugs of interest.
‡Includes patients treated with two or more of the drugs of interest, in any combination/sequence.
§Includes patients admitted to hospital up to and including 10 November 2021.

in their use over time. Of the three drugs of interest, dexamethasone was most widely used in patients hospitalised with COVID-19 in Scotland; while 76.2% of all treated patients received this steroid, only 1.7% and 3.6% of the patients were treated with remdesivir and tocilizumab, respectively. Furthermore, 18.5% of the patients received varying combinations of these three drugs. Although study designs and analytical methods differed widely, use of dexamethasone was also found to be high in other studies: an evaluation of the adoption of corticosteroid treatment in the UK following the RECOVERY trial found that 75.2% of the patients on supplementary oxygen between June 2020 and April 2021 received corticosteroids[30]; an international, multicentre study reported that 68.5% of the patients admitted to ICU were treated with steroids[31]; and among patients hospitalised in Pakistan, the rate was 93.9% for steroids overall and 91.2% for dexamethasone specifically—although these included patients who received combinations of steroids and other drugs.[32] Conversely, treatment with dexamethasone was considerably lower among hospitalised patients who received treatment for COVID-19 in a study conducted in the USA, with only 35.4%; nevertheless, in this study, another 36.6% of the patients received various drug combinations that included dexamethasone.[33] In contrast, use of remdesivir was found to be much higher in other countries than in Scotland, particularly in Pakistan with 45.0% of the hospitalised patients[32] and the USA with 11.5% of the treated patients with remdesivir alone and another 26.0% of patients with various combinations containing remdesivir.[33]

While many underlying reasons may have contributed to these differences in treatment, two conceivable explanations relate to the use of clinical guidelines; and the general availability of drugs. For instance, in the UK, treatment guidelines—although not strictly speaking mandatory—will usually be followed, which may have contributed to the low use of remdesivir where restrictions have been put in place for its use based on time since symptom onset and renal and liver function[34]; in contrast, some shortages in the supply of dexamethasone might have impacted its use in the USA.[16] Of note: drug combinations reported in other studies comprised a much wider range than in our study, and included, for example, azithromycin; hydroxychloroquine; and/or ivermectin—none of which have been authorised for use in COVID-19 in the UK.

Dexamethasone was used throughout the study period, with two prominent peaks: first in January 2021, and then again in September 2021—in line with COVID-19 waves in the UK. In contrast, remdesivir was prescribed early on (it was introduced in guidelines in late May 2020), but its use decreased soon after, being replaced by tocilizumab from January 2021 onwards. Observed changes in treatment pattern over time are, however, in line with evolving knowledge and, consequently, changes in treatment guidelines[9] and have also been reported in other studies: an English study, for example, reported a decrease in the use of remdesivir, and an increase in the use of tocilizumab, during their study period.[35] Interestingly, the number of patients receiving more than one of the studied drugs—most prominently, dexamethasone and tocilizumab—increased over time; anecdotally, the addition of dexamethasone to both tocilizumab and remdesivir treatment has been described in other observational studies, mostly from the USA.[15 33] Since there was wide variety in both the combination and sequencing of drugs and treatment decisions might have been influenced by a number of factors, including disease severity and other, unknown, patient-related aspects, interpretation of these findings is very difficult without further information and would, at this point, be merely speculative.

Duration of treatment was mostly in line with guidelines, with a median of 5 days for remdesivir and a single dose of tocilizumab; similar findings regarding remdesivir have been described in a study conducted in Hong Kong.[36] However, duration of dexamethasone treatment appeared shorter than recommended, with a median of 5 instead of 10 days—although with a rather wide CI of 2–9 days. While differences in median duration of treatment depended on whether patients were ventilated in a study from the USA,[16] reasons underlying our findings are unclear, but could potentially be related to patients being moved to a different ward with no HEPMA coverage (eg, ICU due to deterioration of condition); or being discharged. Alternatively, patients might indeed have stopped treatment with dexamethasone earlier than recommended. Unfortunately, reasons for neither initiating nor discontinuing treatment are recorded on HEPMA.

In Scotland, the majority of patients being treated with any of the three drugs of interest while admitted to hospital for COVID-19, as captured by the available in-hospital prescribing data, were elderly, with a median age of 64 years; furthermore, prescribing data from community care indicated that patients had a range of existing comorbidities, including diabetes, asthma/chronic obstructive pulmonary disease, hypertension, and possibly other cardiovascular diseases. This is in line with expectations based on existing evidence, as disease severity—and, consequently, hospitalisation rates and/or disease outcomes—were shown to be associated with patient age and the presence of certain conditions.[37] Since drug use is closely linked with age (generally speaking, the older a patient, the more medications they will be on) the high level of polypharmacy observed among the study cohort was not entirely unexpected, even though the extent of excessive polypharmacy—that is, 10 or more different drugs—was somewhat surprising; this might, however, have been influenced by the method of analysis, as the number of drugs prescribed to patients reflects both chronic and acute medication and could have been inflated by not accounting for treatment changes (eg, patients switching from one antihypertensive drug to another during the assessed time period).

As an aside, while the majority of the patients captured in this study (73.0%) were unvaccinated, this might primarily be ascribable to the timing of the study, since a large part of the cohort stems from before the vaccination campaign was rolled-out across Scotland. Vaccinations, initially among NHS and care home staff and care home residents, commenced on 8 December 2020, and were subsequently offered in stages; first to clinically vulnerable populations and the elderly, before being available to all adult residents and, subsequently, children aged 5 years and above.[38]

When interpreting results, two further relevant features of prescribing practices need to be taken into account. First, dexamethasone is not the only steroid to be used in patients with COVID-19 (prednisolone is, eg, used in pregnant patients) and, due to intermittent drug shortages, tocilizumab has on occasion been replaced by sarilumab (among others). As these alternatives were not included in the analyses, total drug use may have been underestimated within the cohort. Second, the three drugs of interest are given to different groups of patients, at different points in time on their disease journey. Accordingly, direct comparison between the use of the different drugs, the patients receiving them, and—most importantly—observed patient outcomes, may not be particularly meaningful without considering the complexity of cases and possible variations in treatment based on patient, prescriber and/or wider contextual aspects.

## Study limitations

Although the use of HEPMA offers unprecedented opportunities to conduct drug utilisation research within a secondary care setting, this study also has a number of limitations, which need to be considered when interpreting findings. Most importantly, data coverage was variable between as well as within Health Boards. Some of the major hospitals in NHS Lothian were, for instance, not captured, and data from ICUs were missing as these do not routinely use HEPMA. This may have affected the accuracy of findings; for instance, the number of patients treated with the medicines of interest—in particular, tocilizumab—might have been underestimated due to its use in ICUs. In addition, there are implications on the comparability of findings. Direct comparison of treatment patterns and patient characteristics across Scottish Health Boards, for example, is inadvisable at this point due to the incremental roll-out of the system. Furthermore, although there were clear eligibility and prescribing criteria in place for all three drugs included in this study, we were not able to assess eligibility for treatment; this should, however, not have impacted our findings since the study aim was to describe patterns of use in clinical practice. Finally, caution is advised when interpreting patient characteristics. This study presents a snapshot of patients hospitalised with COVID-19, describing only a sample of patients admitted to hospital who started treatment with any of the drugs of interest in areas where HEPMA has been implemented. Consequently, these characteristics should not be considered representative of all patients hospitalised with COVID-19 in Scotland.

## CONCLUSION

In conclusion, the use of adjuvant medicines in patients hospitalised with COVID-19 in Scotland appears in line with evolving evidence and changing treatment guidelines; nevertheless, findings with regards to the use of multiple drugs and the duration of dexamethasone treatment require further investigation.

## Author affiliations
[1]Strathclyde Institute of Pharmacy and Biomedical Sciences, University of Strathclyde, Glasgow, UK

$^2$Clinical and Protecting Health Directorate, Public Health Scotland Glasgow Office, Glasgow, UK
$^3$Department of Pharmacology and Toxicology, College of Pharmacy, Hawler Medical University, Erbil, Iraq
$^4$NHS Digital, Leeds, UK
$^5$NHS Fife, Kirkcaldy, UK
$^6$Clinical and Protecting Health Directorate, Public Health Scotland, Edinburgh, UK
$^7$The University of Edinburgh Usher Institute of Population Health Sciences and Informatics, Edinburgh, UK
$^8$BREATHE Hub, HDR UK, Edinburgh, UK

**Acknowledgements** We thank Wendy Inglis-Humphrey and Vicky Hammersley for their support with project management and administration. This work is only possible because of the wealth of information collected by the National Health Service as part of routine care.

**Collaborators** EAVE II Collaboration: Colin R Simpson, School of Health, Victoria University of Wellington, Wellington, New Zealand and Usher Institute, University of Edinburgh, Edinburgh, UK; Holly Tibble, Usher Institute, University of Edinburgh, Edinburgh, UK; Chris Robertson, Public Health Scotland, Glasgow, UK and Department of Mathematics and Statistics, University of Strathclyde, UK; critically reviewed the study proposal and the draft manuscript.

**Contributors** TM, AK, SM and MB conceptualised and designed the study. TM, AK and EH curated the data and performed the analyses; IB, JW and AG performed additional analyses. NP, EP, SM, MB and AS contributed to data interpretation. MB and AS had the clinical and academic oversight of the project. All authors have contributed to drafting the paper or revising it critically and have approved of the final version.

**Funding** This study is part of the EAVE II project. EAVE II is funded by the Medical Research Council (MC_PC_19075) with the support of BREATHE - The Health Data Research Hub for Respiratory Health (MC_PC_19004), which in turn is funded through the UK Research and Innovation Industrial Strategy Challenge Fund and delivered through Health Data Research UK. Additional support has been provided through the Scottish Government DG Health and Social Care and the Chief Scientist's Office of the Scottish Government (no grant/award number).

**Competing interests** AS reports grants from NIHR, MRC and HDR-UK; and MB, AK and TM report grants from HDR-UK during the conduct of this study.

**Patient and public involvement** Patients and/or the public were involved in the design, or conduct, or reporting, or dissemination plans of this research. Refer to the Methods section for further details.

**Patient consent for publication** Not applicable.

**Ethics approval** Ethical and information governance approvals have been obtained by the National Research Ethics Service Committee (REC), South East Scotland 02 (REC number: 12/SS/0102) and the Public Benefit and Privacy Panel for Health and Social Care (reference number: 1920-0279), respectively. Study data were anonymised prior to being made available for analyses; and held within a secure environment with restrictions on access. The need for consent was waived by the ethics committee. Informed consent was not required due to the nature of this study, using anonymised data routinely collected in clinical practice. No identifiable information has been used for analyses.

**Provenance and peer review** Not commissioned; externally peer reviewed.

**Data availability statement** National Health Service data are confidential and only available upon request subject to approval by a Caldicott Guardian/the Public Benefit and Privacy Panel for Health and Social Care in Scotland, and the Data Access Request Service in England.

**ORCID iDs**
Tanja Mueller http://orcid.org/0000-0002-0418-4789
Amanj Kurdi http://orcid.org/0000-0001-5036-1988
Stuart McTaggart http://orcid.org/0000-0001-6060-9019
Marion Bennie http://orcid.org/0000-0002-4046-629X

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
