## [Reviewer comments · BMJ Open]

ARTICLE DETAILS

TITLE (PROVISIONAL)	Assessing medication use patterns in patients hospitalised with COVID-19: a retrospective study
AUTHORS	Mueller, Tanja; Kurdi, Amanj; Hall, Elliott; Bullard, Ian; Wapshott, Jo; Goodfellow, Anna; Platt, Niketa; Proud, Euan; McTaggart, Stuart; Bennie, Marion; Sheikh, Aziz

VERSION 1 – REVIEW

REVIEWER	Carlo Piccinni Fondazione ReS (Ricerca e Salute) - Research and Health Foundation - CINECA partner
REVIEW RETURNED	02-Jul-2022

GENERAL COMMENTS	Dear Editor, I read with great interest the article by Mueller and colleagues entitled "Assessing medication use patterns in patients hospitalized with COVID-19: a retrospective study", proposed as an original research article for BMJ Open. This is a descriptive study focused on the Drugutilisation (DU) of 3 drugs (i.e. remdesivir, dexamethasone and tocilizumab) in patients hospitalized for COVID-19 in Scotland. The topic of this article is of interest because DU studies are important to better understand the actual use of medicines, the impact of events on them, and, finally, to design future governance measures. The STROBE list reported by the authors resulted useful to better referee the article. I very much appreciated the details on the pattern of drug use reported in the results section I consider the core of this study and that should be further valorized by the authors. However, I have some major concerns on this article: 1. Although the aim of this study is clear (i.e. to describe the pattern of use of 3 drugs in hospitalized COVID-19 patients) the conclusions reported by the authors are out of this objective. As matter of fact, the authors concluded underling the value and usefulness of two analyzed databases (i.e. HEPMA/EPMA), while, in the abstract, the conclusion paragraph is more in line with the aim of the study. Therefore, the main concern of this paper is the scarce coherence among different sections (especially, the aim, methods, results, and conclusion). In particular, the conclusions are generic and linked to the methods and not to the results. Moreover, the advantages reported for the use of databases are well-known in other countries (such as Italy) where "real-world data" (RWD) are used until more time and are widely used for Pharmacoepidemiological (including DU) studies.2. Another concern is due to the absence of comparison on results with other similar studies, also performed in different countries
---

	where RWD are largely used and where the problems arising in the conclusions are already resolved. 3. Considering the “very small” sample of the studied cohort, the incompleteness of data (as declared by authors at line 297), and the absence of a denominator (i.e. the entire group of COVID-19 hospitalized patients), this study appears not representative of no part of the population. Therefore, the utility of this DU study is compromised. Also, minor concerns are present: 1. Have the authors used the ATC classification to analyze drugs? If yes, they should declare it and specify the used version of ATC. 2. The linkage of different databases should be able to provide more clinical characteristics of the studied cohort. In this study, few characteristics were reported. For example, no information on comorbidities was provided, although this information could be derived from the hospitalization database, or from the pharmaceutical database (by using proxies). The authors, moreover, could consider the use of complexity scores (e.g. Charlson index, CDS, etc) based on their databases to better describe the clinical status of their cohort. 3. Finally, is not clear the usefulness to report results coming from the replication of the analysis on England databases. These last resulted in smaller and more problematic than Scotland ones. For all these reasons I suggest to reject this work or to require a complete revision of it.
--	--

REVIEWER	Hemalkumar Mehta Johns Hopkins University Bloomberg School of Public Health, Epidmeiology
REVIEW RETURNED	02-Aug-2022

GENERAL COMMENTS	The authors evaluated medication use patterns (remdesivir, dexamethasone and tocilizumab) in hospitalized COVID-19 patients in Scotland from March 2020 through November 2021. Dexamethasone monotherapy (76%) was most commonly used. Treatment patterns changed over time. The authors concluded that the use of adjuvant medicine in hospitalized patients appears in line with evolving evidence and changing treatment guidelines. This is a descriptive study with straightforward analysis. It is important to understand the utilization of drugs in Scotland in the first 2 years of the pandemic. Authors need to contextualize their analysis and findings in relation to the key regulator events and clinical guidelines – which is lacking in the current paper. Also, some work is needed to address two methods issues - selection of the cohort (hospitalized individuals with COVID-19, no just those who use one of the three drugs) and reporting of findings (percentage of drug use in trends analysis rather than number of patients using drugs). Finally, their conclusion should directly relate to the objective. I have several questions and suggestions to improve the paper. 1. Introduction: I think the introduction can be shortened and sharpened a bit. For example, you may remove the first paragraph and remove the first line of the second paragraph about vaccine. You can start by describing the burden of COVID-19 in Scotland and what pharmacological treatments were available in the first two years of the pandemic (what you have in the second paragraph). 2. Introduction (line 25): Add, “in-hospital drug use IN SCOTLAND”. There are studies in the US, for example - https://www.acpjournals.org/doi/10.7326/M21-0857. 3. Introduction: I am not sure what’s the relevance of the fourth
--

	paragraph in this study (lines 32-42). This can be perhaps removed or considerably shortened if authors feel it's relevant. Alternatively, some of this information can be moved in methods. 4. Introduction: How treatment evidence changed, drugs were approved, or guidelines evolved in Scotland for these three drugs – these are important points and should be emphasized in the introduction. 5. Study population: Rather than including individuals who received at least one of the three drugs while being admitted to the hospital, a better approach would be to select all hospitalized individuals with COVID-19. Then, look at medication use among these populations. In the current approach, you would exclude people who were hospitalized but did not receive any of these three drugs from your study. This will artificially increase % use of drug use. 6. Study outcomes: I would relabel this as drug use in lines 91, 92 and other places. 7. Partial replication of analyses in England: This is mentioned in methods and results. I don't quite follow the usefulness of this section. Why did you do this analysis? Explain how this analysis supplements or strengthens your primary analysis. Alternatively, if this is not important then remove this section. 8. Statistical analysis: You need to describe in detail your statistical analysis section. You can describe that you did trends in relation to key events, and calculated drug use by mono vs. combination therapy. You need to describe how you calculated the percentage of drug use in each month. For example, how you treated an individual whose hospital stay spanned over two months, for example, admitted in the end of October and discharged in early November. 9. Figure 1 needs some work. Relabel X-axis to ensure the reader knows what those numbers are (months/year). You need to have the percentage of patients on Y-axis, not the number of patients. Also, add key regulatory or clinical guideline events or COVID-19 peak waves to the figure to put things in perspective. You can also show when vaccines were approved for public use. 10. Figure 1. I am not sure what's the reason for the drop in drug use in April-May 2021. Is it a function of a lower number of hospitalizations in that time? You need a percentage of drug use on Y-axis to account for changing hospitalization rates over time in Scotland. 11. Results (lines 123-162): You need to relate increasing or decreasing use with key events. Even if you say use varied with COVID-19 waves, you need to mention what waves from what-to-what time period. 12. Results (lines 164-170): You don't introduce this topic in the introduction and as a reader, I am wondering why you looked at mono vs. combination therapy. Are there any guideline recommendations about the use of combination therapy? What's the significance of this whole analysis? Please clarify. 13. Results (lines 178-209): You can perhaps remove hospital admission-related results from this section and put them in the very first paragraph of the results (page 7). This section can describe treatment duration and initiation for three drugs only. Perhaps relabel the heading. 14. Discussion: Your first line in the discussion states "Using routinely collected data from clinical practice, our findings demonstrate changes in the use of specific drugs over time, aligned with waves of COVID-19 infections and changing treatment guidelines." But, your results did not demonstrate these findings. For instance, Figure 1 did not show how treatment use changed w.r.t. covid waves or changing treatment guidelines. Either clearly show
--	---

	this in results or reframe this sentence so that it is supported by your results. This comment also applies to abstract and manuscript conclusions. 15. Discussion: You probably need to reorganize your discussion and highlight (i) how your results compare with prior studies, (ii) how your results provide new information which is clinically or policy-relevant (was drug use appropriate, inappropriate, less etc.). You can follow BMJ guidelines for the discussion to better organize and write this section. 16. Discussion: I think you can remove the second paragraph from the discussion or considerably shorten it and include it somewhere. 17. Discussion: Why remdesivir use was so low? I think the use is higher in US than what you found in Scotland (https://www.acpjournals.org/doi/10.7326/M21-0857). Are there any studies on remdesivir use Europe/UK that you can compare? You can discuss this. 18. Conclusions: Is this study about showing the usefulness of HEPMA/EPMA? No. So, I would remove that from the conclusion and focus on your aim – describing drug use among COVID-19 patients. It's ok to mention HEPMA/EPMA in discussion but not in conclusions.
--	---

VERSION 1 – AUTHOR RESPONSE

Reviewer 1: Dr Carlo Piccinni

1. Although the aim of this study is clear (i.e. to describe the pattern of use of 3 drugs in hospitalized COVID-19 patients) the conclusions reported by the authors are out of this objective. As matter of fact, the authors concluded underling the value and usefulness of two analyzed databases (i.e. HEPMA/EPMA), while, in the abstract, the conclusion paragraph is more in line with the aim of the study. Therefore, the main concern of this paper is the scarce coherence among different sections (especially, the aim, methods, results, and conclusion). In particular, the conclusions are generic and linked to the methods and not to the results. Moreover, the advantages reported for the use of databases are well-known in other countries (such as Italy) where “real-world data” (RWD) are used until more time and are widely used for Pharmacoepidemiological (including DU) studies.

Response: *We thank the reviewer for highlighting discrepancies between the different sections of our manuscript. To rectify this, we have thoroughly re-assessed all sections and made a number of amendments throughout to better align the different parts of the manuscript. Most importantly, we have revised the conclusion to be linked to the results, not the methods:*

“In conclusion, the use of adjuvant medicines in hospitalised COVID-19 patients in Scotland appears in line with evolving evidence and changing treatment guidelines; nevertheless, findings with regards to the use of multiple drugs and the duration of dexamethasone treatment require further investigation.” (lines 316 – 319)

2. Another concern is due to the absence of comparison on results with other similar studies, also performed in different countries where RWD are largely used and where the problems arising in the conclusions are already resolved.

Response: *Thank you for pointing this out. We have expanded the discussion section accordingly and added further references to provide a comparison with studies performed in other settings, which are listed at the end of this document:*

“Although study designs and analytical methods differed widely, use of dexamethasone was also found to be high in other studies: an evaluation of the adoption of corticosteroid treatment in the UK following the RECOVERY trial found that 75.2% of patients on supplementary oxygen between June 2020 and April 2021 received corticosteroids (Nähri 2022); an international, multicentre study reported that 68.5% of patients admitted to ICU were treated with steroids (Amer 2021); and among patients hospitalised in Pakistan, the rate was 93.9% for steroids overall, and 91.2% for dexamethasone specifically – although these included patients who received combinations of steroids and other drugs (Mustafa 2022). Conversely, treatment with dexamethasone was considerably lower among hospitalised patients who received treatment for COVID-19 in a study conducted in the US, with only 35.4%; nevertheless, in this study, another 36.6% of patients received various drug combinations that included dexamethasone (Ayodele 2021). In contrast, use of remdesivir was found to be much higher in other countries than in Scotland, particularly in Pakistan with 45.0% of hospitalised patients (Mustafa 2022), and the US with 11.5% of treated patients with remdesivir alone and another 26.0% of patients with various combinations containing remdesivir (Ayodele 2021).” (lines 209 – 224)

“Observed changes in treatment pattern over time are, however, in line with evolving knowledge and, consequently, changes in treatment guidelines (MHRA 2022), and have also been reported in other studies: an English study for example reported a decrease in the use of remdesivir, and an increase in the use of tocilizumab, during their study period (Freeman 2022). Interestingly, the number of patients receiving more than one of the studied drugs – most prominently, dexamethasone and tocilizumab – increased over time; anecdotally, the addition of dexamethasone to both tocilizumab and remdesivir treatment has been described in other observational studies, mostly from the US (e.g., Gupta 2021 , Ayodele 2021).” (lines 239 – 246)

“Duration of treatment was mostly in line with guidelines, with a median of five days for remdesivir and a single dose of tocilizumab; similar findings regarding remdesivir have been described in a study conducted in Hong Kong (Wong 2021). However, duration of dexamethasone treatment appeared shorter than recommended, with a median of five instead of 10 days – albeit with a rather wide confidence interval of two to nine days. While differences in median duration of treatment depended on whether patients were ventilated in a study from the US (Mehta 2021), reasons underlying our findings are unclear, but could potentially be related to patients being

moved to a different ward with no HEPMA coverage (e.g., ICU due to deterioration of condition); or being discharged.” (lines 251 – 259)

3. Considering the “very small” sample of the studied cohort, the incompleteness of data (as declared by authors at line 297), and the absence of a denominator (i.e. the entire group of COVID-19 hospitalized patients), this study appears not representative of no part of the population. Therefore, the utility of this DU study is compromised.

Response: *We acknowledge the limitations of our study, particularly with regards to the limited sample size and the variability in data coverage across the different regions in Scotland, and appreciate that our findings may not be entirely representative of the Scottish population; these limitations have explicitly been stated within the text. Nevertheless, we consider our study useful since the data presented therein offer previously unknown insights into clinical practice at the time of study conduct, and as such do contribute to the slowly growing body of evidence – particularly since our findings comprise aspects such as duration of treatment, reports of which are still scarce.*

With regards to the absence of a denominator: the aim of our study was to describe the pattern of use of the three drugs of interest in hospitalised patients with COVID-19; hence, the study population comprises – by definition – patients who received any of these drugs. Restricting the study cohort to only those patients who were exposed to the drug of interest is not unusual and has also been done in other COVID-19 drug utilisation studies (e.g., Ayodele 2021, Tejada 2022). Although we certainly agree that including patients who did not receive any treatment is useful as a comparator group when aiming to answer certain other questions especially relating to effectiveness and safety, these were outwith the scope of our study. The included drugs had clear eligibility and prescribing criteria, with those patients not receiving treatment most likely being ineligible for various reasons; unfortunately, we were not able to assess eligibility for treatment. This is now more clearly stated in the context of discussing study limitations.

4. Have the authors used the ATC classification to analyze drugs? If yes, they should declare it and specify the used version of ATC.

Response: *The Hospital Electronic Prescribing and Medicines Administration system data is indexed using the NHS “Dictionary of Medicines and Devices” (dm+d), a terminology developed by NHS England and related to SNOMED-CT (<https://www.nhsbsa.nhs.uk/pharmacies-gp-practices-and-appliance-contractors/dictionary-medicines-and-devices-dmd>). Drugs of interest were therefore identified using the appropriate dm+d descriptions; we have not used the ATC system to analyse drugs for this study. Specifics of the methodology, including a description of all variables, are available in the published protocol which has been referred to in the methods section of the manuscript (doi: 10.1136/bmjopen-2021-054861).*

5. The linkage of different databases should be able to provide more clinical characteristics of the studied cohort. In this study, few characteristics were reported. For example, no information on comorbidities was provided, although this information could be derived from the hospitalization database, or from the pharmaceutical database (by using proxies). The authors, moreover, could consider the use of complexity scores (e.g. Charlson index, CDS, etc) based on their databases to better describe the clinical status of their cohort.

Response: *Some information on comorbidities, using prescription drugs as a proxy, were already included in the baseline characteristics; we have now expanded on this. Furthermore, we have added a Charlson score as suggested (table 1):*

“While approximately half of all patients (n=2159, 53.1%) had a Charlson score of 0 indicating no comorbidities requiring hospitalisation during the five-year period prior to their COVID-19 admission the majority (n=3616, 89.0%) had received at least one prescription drug in primary care in the preceding six months. The most commonly prescribed medication included antihypertensive drugs, statins, and antithrombotic drugs; polypharmacy – i.e. being prescribed five or more different medications simultaneously – was observed among 2752 patients (67.7%).” (lines 134 – 140)

6. Finally, it is not clear the usefulness to report results coming from the replication of the analysis on England databases. These last resulted in smaller and more problematic than Scotland ones.

Response: *We appreciate that adding the partial replication of analyses based on English data may not have added further relevant insights, particularly because of its limited sample size. Therefore, we have removed this part from the manuscript.*

Reviewer 2: Dr Hemalkumar Mehta

1. Introduction: I think the introduction can be shortened and sharpened a bit. For example, you may remove the first paragraph and remove the first line of the second paragraph about vaccine. You can start by describing the burden of COVID-19 in Scotland and what pharmacological treatments were available in the first two years of the pandemic (what you have in the second paragraph).

Response: *Thank you for this suggestion. We have modified the text to better focus the introduction as follows: removed the first paragraph; shortened the sentence on vaccines; and added a new introductory sentence describing the burden of COVID-19 in Scotland:*

“Similar to other countries globally, COVID-19 had a considerable impact on population health in the UK; in Scotland alone, approximately 44,300 hospital admissions and 12,500 deaths have been attributed to the disease between March 2020 and December 2021. Even though vaccines were rolled-out at speed, the search for effective treatments is ongoing [...]” (lines 2 – 6)

2. Introduction (line 25): Add, “in-hospital drug use IN SCOTLAND”. There are studies in the US, for example Mehta et al (2021) <https://doi.org/10.7326/M21-0857>.

Response: We have added “in Scotland” to the sentence as suggested (line 29). We have also amended subsequent wording and added the provided reference for clarification as follows:

“Although there are exceptions (Mehta 2021), studies rarely provide in-depth information on treatment patterns (including temporal and geographical variation); details on duration of treatment are even less common, possibly due to the unavailability of in-depth, patient-level information.” (lines 32 – 35)

3. Introduction: I am not sure what’s the relevance of the fourth paragraph in this study (lines 32-42). This can be perhaps removed or considerably shortened if authors feel it’s relevant. Alternatively, some of this information can be moved in methods.

Response: This section was included to offer further context of the wider study setting with regards to data availability; as such, we consider this information relevant. Nevertheless, we have as suggested shortened the paragraph to remove redundant text.

4. Introduction: How treatment evidence changed, drugs were approved, or guidelines evolved in Scotland for these three drugs – these are important points and should be emphasized in the introduction.

Response: We agree that these are important aspects, and have expanded on this in the introduction as follows:

“Medicines recommended for use in patients hospitalised with COVID-19 in the UK at the time of study conduct comprised remdesivir, an antiviral drug (introduced in guidelines 29th May 2020); steroids, most prominently dexamethasone (16th June 2020); and monoclonal antibodies including tocilizumab (8th January 2021). Guidelines have however undergone frequent changes over the last two years (MHRA 2022). Dexamethasone is now routinely given to patients with low oxygen saturation, primarily because it is an effective and generally well tolerated anti-inflammatory drug, with extensive experience in usage. Remdesivir is recommended in patients receiving low-flow additional oxygen, depending on renal and liver function, to be given within the first few days of infection as it is deemed inefficient at later stages; tocilizumab is recommended in patients receiving respiratory support (NHS England 2022).” (lines 11 – 21)

5. Study population: Rather than including individuals who received at least one of the three drugs while being admitted to the hospital, a better approach would be to select all hospitalized individuals with COVID-19. Then, look at medication use among these populations. In the current approach, you would exclude people who were hospitalized but did not receive any of these three drugs from your study. This will artificially increase % use of drug use.

Response: *The aim of our study was to describe the pattern of use of the three drugs of interest in hospitalised COVID-19 patients treated with these drugs; hence, the study population comprises – by definition – hospitalised COVID-19 patients who received any of these drugs. Our aim was not to estimate the prevalence of drug use among hospitalised COVID-19 patients in general. Restricting the study cohort to only those patients who were exposed to the drug of interest is not unusual and has also been done in other COVID-19 drug utilisation studies (e.g., Ayodele 2021, Tejada 2022). Although we certainly agree that including patients who did not receive any treatment is useful as a comparator group when aiming to answer certain other questions especially relating to effectiveness and safety, these were outwith the scope of our study, but may be the focus of future work. The included drugs had clear eligibility and prescribing criteria, with those patients not receiving treatment most likely being ineligible for various reasons; unfortunately, we were not able to assess eligibility for treatment. Treatment eligibility and use is certainly another area of possible follow-up work.*

6. Study outcomes: I would relabel this as drug use in lines 91, 92 and other places.

Response: *We have relabelled the section accordingly: “Drug use, covariates, and statistical analyses” (line 88)*

7. Partial replication of analyses in England: This is mentioned in methods and results. I don't quite follow the usefulness of this section. Why did you do this analysis? Explain how this analysis supplements or strengthens your primary analysis. Alternatively, if this is not important then remove this section.

Response: *We appreciate that adding the partial replication of analyses based on English data may not have added further relevant insights, particularly because of its limited sample size. Therefore, we have removed this part from the manuscript.*

8. Statistical analysis: You need to describe in detail your statistical analysis section. You can describe that you did trends in relation to key events, and calculated drug use by mono vs. combination therapy. You need to describe how you calculated the percentage of drug use in each month. For example, how you treated an individual whose hospital stay spanned over two months, for example, admitted in the end of October and discharged in early November.

Response: *A patient who was admitted at the end of October and immediately (within the same month) started treatment with one of the drugs of interest would have been included in the count for the drug they were started on for the month of October, regardless of when they were discharged; in contrast, a patient who was admitted at the last day of October but started treatment the first day in November would have been included in the count for November. For patients who received more than one drug, the date of the first recorded prescription was used for the purpose of categorising treatment over time.*

We have now expanded the statistical methods section to provide a fuller description of how we categorised patient numbers and percentages over time:

“Patients’ index dates – i.e., the date of the first recorded prescription for any of the drugs of interest for each patient – were used to classify treatment over time.” (lines 100 – 101)

9. Figure 1 needs some work. Relabel X-axis to ensure the reader knows what those numbers are (months/year). You need to have the percentage of patients on Y-axis, not the number of patients. Also, add key regulatory or clinical guideline events or COVID-19 peak waves to the figure to put things in perspective. You can also show when vaccines were approved for public use.

Response: *We have relabelled figure 1 as suggested. To provide additional context, we have added a graph showing the number of COVID-19 hospitalisations during the study period (figure 2), to replace the figure with confirmed number of cases originally included in the supplementary material. Figure 1 shows the number of patients to facilitate comparison between the observed trends in treatment and underlying trends in hospitalisations; as our study cohort does not include an untreated cohort, changing the y-axis to percentages may give a misleading picture. We have added a figure representing proportional changes in drug use over time to the supplementary material instead, to highlight the changes in treatment during the study period.*

10. Figure 1. I am not sure what’s the reason for the drop in drug use in April-May 2021. Is it a function of a lower number of hospitalizations in that time? You need a percentage of drug use on Y-axis to account for changing hospitalization rates over time in Scotland.

Response: *The drop in use in April/May 2021 is in line with a decline in hospitalisations in Scotland during this time, which would have had an impact on the population eligible for treatment. To provide context for the findings, we have added a graph showing the number of COVID-19 hospitalisations during the study period (figure 2).*

11. Results (lines 123-162): You need to relate increasing or decreasing use with key events. Even if you say use varied with COVID-19 waves, you need to mention what waves from what-to-what time period.

Response: *Thank you for highlighting this – we have added a graph showing the number of COVID-19 hospitalisations during the study period to provide better context (figure 2).*

12. Results (lines 164-170): You don’t introduce this topic in the introduction and as a reader, I am wondering why you looked at mono vs. combination therapy. Are there any guideline recommendations about the use of combination therapy? What’s the significance of this whole analysis? Please clarify.

Response: *In the introduction, we referred to the relevant treatment guidelines in general terms, and we appreciate that the possibility of combination treatment was not explicitly mentioned. The main objective of this study was to describe observed patterns of medication use among hospitalised patients with COVID-19, this naturally includes patients receiving more than one of the included drugs (either concomitantly or consecutively) if that is what has been done in practice. This has been alluded to in the study population section (“[...] received at least one of the medications of interest [...]”). To clarify, we have expanded on this in both the introduction and the methods section:*

“While usually given separately, consecutive (e.g., dexamethasone together with tocilizumab) or subsequent treatment with multiple drugs can be considered based on clinical condition (NHS England 2021, NHS England 2022).” (lines 21 – 23)

“The primary outcome was observed treatment patterns among patients with any of the medications of interest over the duration of the study period, including both monotherapy as well as treatment with any combinations thereof.” (lines 89 – 91)

13. Results (lines 178-209): You can perhaps remove hospital admission-related results from this section and put them in the very first paragraph of the results (page 7). This section can describe treatment duration and initiation for three drugs only. Perhaps relabel the heading.

Response: *As suggested, we have moved the hospital admission-related details to the first paragraph of the results section. The section has been relabelled “Treatment episodes”.*

14. Discussion: Your first line in the discussion states “Using routinely collected data from clinical practice, our findings demonstrate changes in the use of specific drugs over time, aligned with waves of COVID-19 infections and changing treatment guidelines.” But, your results did not demonstrate these findings. For instance, Figure 1 did not show how treatment use changed w.r.t. covid waves or changing treatment guidelines. Either clearly show this in results or reframe this sentence so that it is supported by your results. This comment also applies to abstract and manuscript conclusions.

Response: *Thank you for pointing this out. In addition to amending the figures within the text to provide better context (see also previous comments), we have revised this sentence as follows:*

“Using routinely collected data, our findings provide a picture of clinical practice during the study period in Scotland and demonstrate differences in the use of specific drugs for the treatment of COVID-19, as well as changes in their use over time.” (lines 203 – 205)

15. Discussion: You probably need to reorganize your discussion and highlight (i) how your results compare with prior studies, (ii) how your results provide new information which is clinically or

policy-relevant (was drug use appropriate, inappropriate, less etc.). You can follow BMJ guidelines for the discussion to better organize and write this section.

Response: *We have reorganised the discussion in line with the reviewer’s suggestions.*

16. Discussion: I think you can remove the second paragraph from the discussion or considerably shorten it and include it somewhere.

Response: *We have shortened and repositioned this paragraph within the discussion section as advised by the reviewer.*

17. Discussion: Why remdesivir use was so low? I think the use is higher in US than what you found in Scotland; Mehta et al (2021) <https://doi.org/10.7326/M21-0857>. Are there any studies on remdesivir use Europe/UK that you can compare? You can discuss this.

Response: *Thank you for highlighting this; we have expanded as follows:*

“While many underlying reasons may have contributed to these differences in treatment, two conceivable explanations relate to the use of clinical guidelines; and the general availability of drugs. For instance, in the UK, treatment guidelines – although not strictly speaking mandatory – will usually be followed, which may have contributed to the low use of remdesivir where restrictions have been put in place for its use based on time since symptom onset and renal and liver function (Scottish Government, 2022); in contrast, some shortages in the supply of dexamethasone might have impacted its use in the US (Mehta, 2021). Of note: drug combinations reported in other studies comprised a much wider range than in our study, and included, for example, azithromycin; hydroxychloroquine; and/or ivermectin – none of which have been authorised for use in COVID-19 in the UK.” (lines 225 – 234)

18. Conclusions: Is this study about showing the usefulness of HEPMA/EPMA? No. So, I would remove that from the conclusion and focus on your aim – describing drug use among COVID-19 patients. It’s ok to mention HEPMA/EPMA in discussion but not in conclusions.

Response: *We agree that the conclusion in its original form was not in line with the stated study aim; therefore, we have revised this section as follows:*

“In conclusion, the use of adjuvant medicines in hospitalised COVID-19 patients in Scotland appears in line with evolving evidence and changing treatment guidelines; nevertheless, findings with regards to the use of multiple drugs and the duration of dexamethasone treatment require further investigation.” (lines 316 – 319)

VERSION 2 – REVIEW

REVIEWER	Hemalkumar Mehta Johns Hopkins University Bloomberg School of Public Health,
----------	---

	Epidmeiology
REVIEW RETURNED	06-Oct-2022
GENERAL COMMENTS	Great job revising this paper.